# Counterfactual Voting Adjustment for Quality Assessment and Fairer Voting in Online Platforms with Helpfulness Evaluation

Chang Liu[1]   Yixin Wang[2]   Moontae Lee[1][3]

## Abstract

Efficient access to high-quality information is vital for online platforms. To promote more useful information, users not only create new content but also evaluate existing content, often through helpfulness voting. Although aggregated votes help service providers rank their user content, these votes are often biased by disparate accessibility per position and the cascaded influence of prior votes. For a fairer assessment of information quality, we propose the Counterfactual Voting Adjustment (CVA), a causal framework that accounts for the context in which individual votes are cast. Through preliminary and semi-synthetic experiments, we show that CVA effectively models the position and herding biases, accurately recovering the predefined content quality. In a real experiment, we demonstrate that reranking content based on the learned quality by CVA exhibits stronger alignment with both user sentiment and quality evaluation assessed by GPT-4o, outperforming system rankings based on aggregated votes and model-based rerankings without causal inference. Beyond the individual quality inference, our embeddings offer comparative insights into the behavioral dynamics of expert user groups across 120 major StackExchange communities.

## 1. Introduction

User-generated content is influential information for decision-making and knowledge discovery. Customer reviews on merchant products and services are essential elements of electronic Word-Of-Mouth (eWOM) (Babić Rosario et al., 2016), where 97% of customers rely on online reviews for their daily shopping decisions (Murphy, 2017). User answers in Q&A communities help build large-scale knowledge repositories through the wisdom of crowds (Wang et al., 2013). For instance, StackOverflow — the largest iconic community within the StackExchange expert Q&A forums — alone recorded over 24 million questions and 36 million answers (sta). This paper identifies these customer reviews and user answers as key examples of **user-generated responses** provided for various products and questions.

Volume does not always indicate quality. Although user-generated responses are abundant, their sheer size and diversity make it difficult to identify genuinely helpful information. To address this challenge, many service providers frequently implement binary voting systems, allowing users to rate individual responses as helpful or not. However, even when aggregated, these helpfulness votes may fail to accurately capture the true quality of the responses. Due to the limited perception of human users (Joachims et al., 2007), more accessible responses displayed at the top tend to attract more votes, leading to **position bias**. Furthermore, users often conform to majority opinion: the visible count of prior votes can significantly steer subsequent evaluations (Lee et al., 2016), amplifying **herding bias**. Over time, these biases mutually reinforce a cascading effect: favoring content that gains early popularity and overshadowing potentially superior yet less accessible future content.

To fairly assess information quality while mitigating position and herding biases, we should be capable of addressing two counterfactual questions: *What if the content were displayed in a different position*, and *what if the content had an equal number of positive and negative votes?* In interactive systems, however, it is infeasible to create parallel universes where the same content is evenly distributed across all positions with a balanced number of votes. To overcome this challenge, we introduce the **Counterfactual Voting Adjustment (CVA)**, a causal framework that systematically accounts for the contextual factors influencing individual votes. Unlike existing models that rely solely on static data, CVA leverages voting trajectories to simultaneously adjust position and herding biases, offering a principled approach to inferring the causal effect of helpfulness voting on content quality. This framework facilitates fairer rankings that reduce the undue influence of early accessibility and social

[1]Department of Information and Decision Sciences, University of Illinois Chicago, Chicago, United States [2]University of Michigan, Ann Arbor, MI, United States [3]LG AI Research, Seoul, Korea. Correspondence to: Moontae Lee <moontae@uic.edu>.

*Proceedings of the 42$^{nd}$ International Conference on Machine Learning*, Vancouver, Canada. PMLR 267, 2025. Copyright 2025 by the author(s).

conformity, ultimately leading to a reliable identification of high-quality content among user-generated responses.

Extensive validations and analyses confirm the effectiveness of our CVA framework. First, preliminary experiments demonstrate how CVA mitigates position and herding biases, inferring information quality as desired. Second, a semi-synthetic dataset, designed to closely emulate real community statistics, shows that CVA achieves the strongest alignment with prior content quality. Third, since the true quality of information in real StackExchange data is unknown, we leverage GPT-4o, a Large Language Model (LLM), to evaluate sentiment in user comments and content quality by LLM-as-a-judge like human experts (Chiang & Lee, 2023). The results prove that our CVA re-rankings significantly outperform both existing system rankings and model-based re-rankings that lack causal inference. Fourth, community embeddings reveal varying degrees of behavioral biases, generalizing our findings on representative communities across the major StackExchange forums. Finally, through a case study illustrating how CVA answers to the key counterfactual questions, we discuss its potential benefits for service providers as well as users.

## 2. Related Work

Our work expands recent studies in position and herding bias verification, helpfulness voting modeling, and approaches for mitigating biases and inferring information quality.

We aim to simultaneously mitigate position and herding bias and uncover fairer quality estimates of responses rather than just quantify biases and predict the next votes.

**Previous studies proved the existence of the position and herding biases.** Wan (2015) identified the Matthew effect, where early-posted reviews garner more votes due to visibility, and the Ratchet effect, which sustains their dominance through cumulative feedback loops. (Zhou & Guo, 2017; Alzate Barricarte et al., 2024; Sipos et al., 2014) also verified that reviews ranked higher always get higher review visibility. (Risselada et al., 2018; Tseng et al., 2023) verified social conformity and demonstrated herd behavior in online voting systems, revealing that prior votes strongly influence subsequent decisions. Guo et al. (2019) used neurological tools, showing that users experience positive neurological feedback when their votes align with majority opinions, suggesting cognitive biases toward popular reviews. Liu et al. (2007) mixes these two biases into *winner circle bias*.

**Modeling the helpfulness voting.** In the Computer Science field, modern studies predominantly employ machine learning and deep learning techniques to predict the helpfulness voting score using data as a snapshot. Yin et al. (2021) proposed a semi-supervised learning framework utilizing unlabeled reviews. Palahan (2023); Sharma et al. (2023)

provided a comparative analysis of deep learning models, finding that CNN outperforms traditional classifiers, RNNs, and XGBoost. Kastrati et al. (2024) used deep learning to identify helpful reviews and integrated textual and meta-data attributes. Researchers in the business field also leverage computational models to improve the performance of helpfulness prediction. Tay et al. (2020) proposed Dirichlet distribution-based aggregation models. Deng et al. (2020) divided voting processes into initial and cumulative stages, with heuristic and systematic cues influencing these stages differently. Du et al. (2021) proposed an end-to-end neural architecture to capture the missing interaction between reviews and their contextual neighbors. Mitra & Jenamani (2021) proposed multi-perspective models combining lexical, structural, and sequential features. Wang et al. (2021) applied random forests and gradient boosting techniques to analyze helpfulness on the Steam gaming platform. Lee et al. (2021) found that XGBoost outperformed other models for predicting review helpfulness in restaurant data. Wang et al. (2020a) proposed Bayesian ranking techniques that balance early and late-posted reviews, only ensuring fairer visibility distribution without considering herding bias. Dev et al. (2019) proved causal effects of different impression signals, such as aggregate vote thus far and position of content, by adopting an instrumental variable (IV) framework. However, their approach is unable to isolate social influence bias with position bias.

**Mitigating the biases.** The Chinese Voting Process (CVP) proposed a generative model that infers the underlying quality of individual responses by de-biasing the herding bias (Lee et al., 2016). While the CVP utilizes trajectories of voting to re-weight the importance of individual votes per different contexts, this model cannot de-bias the position bias together with the herding bias. Note that it is important to recognize the observational nature of our dataset. Even if we reconstruct all trajectories of writing new responses and voting to existing responses similar to the CVP, our dataset consists only of the votes that have been cast to existing answers. To address this, some studies consider both whether to vote and helpfulness vote ratio as outcomes (Kuan et al., 2015; Wu et al., 2021). They and CVP both proposed a two-stage model that cannot mitigate the position bias and herding bias at the same time. CVP also did a cross-community comparison. Helpfulness votes represent the subjective valuation of the information judged by others, and they are also the aggregated perceived utility of the information (Kuan et al., 2015). Due to its subjectivity, the sensitivities to biases vary among different communities.

To address biases and confounders in causal inference for recommendation studies, exposure bias widely exists due to self-selection through user exposure or ranking policy of the systems itself. Liang et al. (2016a;b); Wang et al. (2020b) model user exposure in recommendation as a latent variable

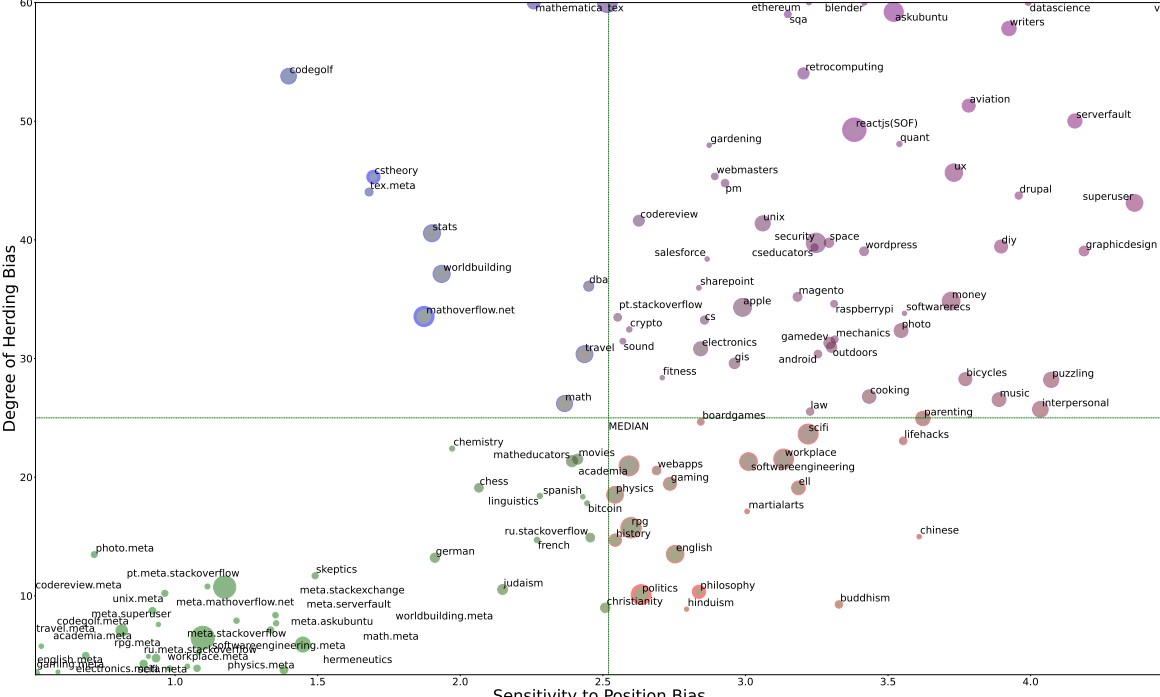

Figure 1: Analyses of different voting dynamics across the 120 largest communities on StackExchange. First, math-oriented communities such as *mathoverflow.net* and *stats* exhibit high conformity but are less influenced by trends due to the exact and provable nature of their knowledge content. Second, computer-related communities like *reactjs(SOF)* and *askubuntu* are particularly responsive to trending and most useful answers because they share up-to-date and practical experiences. Third, users in *meta* communities are minimally conforming and barely sensitive to trending information, rather featuring diverse opinion as purposed. Finally, *politics* and *philosophy* resemble religious communities with their low conformity but a high sensitivity to popular discussions, likely driven by their personal and societal beliefs. Interestingly, *bitcoin* emerges as a kind of new language that uniquely intersects gaming aspects.

and formalize the problem with empirical risk minimization methods to infer the value of exposure from data (Krauth et al., 2022; Mendler-Dünner et al., 2022). Schnabel et al. (2016) followed a similar process but adopted a Bayesian perspective. Both of them fitted the exposure model and action model separately. However, the explicit exposure data they used is hard to obtain.

Existing methods are unable to answer the two counterfactual questions introduced earlier using only observational data. Our CVA addresses this by combining causal and behavioral modules, as described in the next section.

## 3. Counterfactual Voting Adjustment

While helpfulness votes are often correlated with the quality of responses, they suffer from both herding bias and position bias. The same response may receive different helpfulness votes if the voting user sees a different distribution of existing votes; the votes may also differ even if the same response was placed at different positions. How can we mitigate these biases? In this section, we take a causal inference

approach for bias correction and propose an algorithm for a more faithful quality estimation.

Why take a causal approach to response quality estimation? The key observation is that the helpfulness votes of a response are a consequence of both the quality of the response and its other extrinsic factors (i.e., the response location and its existing vote distribution). In this sense, the helpfulness votes will be a more faithful quality estimator if we live in a fictitious world where all responses are placed at the *same* location (e.g., ranked at the top) and share the *same* distribution of existing votes (e.g., an even distribution of positive and negative votes). That is, for each response, we hope to infer how users would vote *counterfactually* if it were ranked at the top and has an even existing distribution. One can thus use this counterfactual vote estimate to mitigate the herding and position bias in response quality estimation.

**Setup and notations.** We begin by describing the interactive voting setup. In interactive voting, users vote on questions sequentially. At each time step, a user decides whether to choose a response to read and vote (or otherwise write a new response) and then decides to vote positively

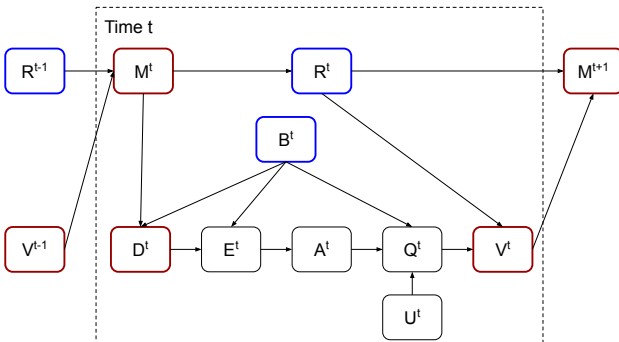

Figure 2: A causal graph for interactive voting that meets our assumptions. At each time step t, observable variables include each answer's existing votes (M), summary statistic of M (R), displayed rank (D), and pre-reading content-related features (B). These affect exposure (unobservable E). If a user chooses to read the answer, post-reading features (unobservable A) are revealed. Together with B and other unobservable factors (U), they shape the perceived quality of the answer (Q), which drives whether it receives a positive or negative vote (V). Over time, current V and R update the future M.

or negatively on the chosen response. Both actions can be affected by the existing votes of each response and its displayed rank. More formally, at time $t$, for question $i$, a user chooses a response from existing responses to work with as the exposure $E_i^t$ (we assume that once a response is chosen, it must be read and voted); we denote the total number of existing responses as $J_i^{t-1}$. This exposure is based on three types of observations. Observation 1 is the existing votes, where we denote $M_{ij}^t, j \in \{1, \ldots, J_i^{t-1}\}$ as the number of positive and negative votes for the $j$th response at time $t$. As a summary statistic of $M$, we further denote the perceived positive vote ratio at time $t$ as $R_{ij}^t$. Observation 2 is the currently displayed rank of the responses, denoted as $D_{ij}^t, j \in \{1, ..., J_{ij}^t\}$, where $D = 1$ means top rank. Observation 3 are some content-related features **before** the responses being actually read which we denoted $B_{ij}^t, j \in \{1, ..., J_{ij}^t\}$.

At time $t$, a user chooses to work with $E_i^t$-th response. She either chooses to read and vote on the $E_i^t = j$th response out of $J_i^{t-1}$ total responses or chooses to write the new response $J_i^{t-1} + 1$. If she chooses to work with an existing response, then she can perceive some other content-related features **after** the response being read, denoted as $A_{ij}^t$. This $A_{ij}^t$ along with $B_{ij}^t$ and other unobservable but independent features $U_t$ will affect the perceived quality of the chosen response at this time ($Q_{ij}^t$) by the user. Finally, this user decides to vote positively on the chosen $E_i^t = j$th response with a probability that depends on the perceived quality of the chosen response at this time $Q_{ij}^t$ and other factors $G_{ij}^t$

via some function $f$,

$$P(V_{ij}^t = 1) = f(Q_{ij}^t, G_{ij}^t),$$

where $G_{ij}^t = (B_{ij}^t, M_{ij}^t, D_{ij}^t)_{j=1}^{J_i^{t-1}}$ includes all other factors that drive the position bias and herding bias. We only consider the relative length of the responses $L_{ij}^t$ as the main factor of $B_{ij}^t$, which is one of our limitations. Our framework can easily incorporate additional factors when they are observable in the future.

Regarding the sequential decision-making of the voting process, the current vote ($V^t$) and current summary of existing votes ($R^t$) will directly affect the next round $M^{t+1}$ at time $t + 1$. Variables in red or blue frames are observable, while those in black frames are unobservable.

**Interactive voting as causal inference.** To estimate the counterfactual votes for each response, we next frame the users' voting behavior as a causal problem at one certain time $t$. The counterfactual votes of interest describe "*what would the votes of the responses be *if* it were displayed at a different rank and/or received different previous votes.*" The treatment is thus the existing votes $M_{ij}^t$ and its display rank $D_{ij}^t$, and the outcome is the voting behavior (positive or negative vote). We denote the *potential vote* of a response as $V_{ij}^t(g_{ij}^t)$ at time $t$ if its existing votes were $m_{ij}^t$ and its display rank were $d_{ij}^t$ while the text stays as is, i.e. $g_{ij}^t = (b_{ij}^t, m_{ij}^t, d_{ij}^t)$.

To perform causal inference for $V_{ij}^t(m, d)$, we rely on a key ignorability condition (Imbens & Rubin, 2015),

$$V_{ij}^t(g_{ij}^t) \perp M_{ij}^t, D_{ij}^t | B_{ij}^t \qquad (1)$$

This assumption says that the only confounder, namely the variable that affects both the extrinsic factors of the responses (e.g., the display rank and the existing votes of responses) and the users' voting behavior, is the response's $B_{ij}^t$ itself. There are no other factors that can affect both simultaneously. This is a common assumption in interactive voting in that a user makes a voting decision based on either the extrinsic factors $M_{ij}^t, D_{ij}^t$ or the response's $B_{ij}$.

This ignorability assumption holds under, for example, the causal graph in Figure 2. (Many other causal graphs also satisfy this assumption.) At the time step $t$, a user decides what to vote next given presentational features before the response is read ($B$), previous votes ($M$), and currently displayed rank ($D$) as input from the system. So $M$ directly decides the presented votes summary $R$, such as positive vote ratio and vote difference. $M$ also affects the displayed rank $D$ at each time since the platform ranks the responses based on vote difference by default. The content-related presentational factor $B$ is a confounder; it affects both the treatments $M$ and $D$, but also the outcome $V$. Moreover, we assume that conditioning on $B$ will block all backdoor

paths (Pearl, 2010), hence satisfying the ignorability assumption Equation (1) which is one of our limitations. Wiping out the effect of $M$ and $D$ on the voting behavior $V$ makes $V$ reflect the true response quality $Q$.

Given this ignorability condition, together with other standard assumptions in causal inference (namely, positivity and stable unit treatment value assumption (Imbens & Rubin, 2015)), one is greenlighted to estimate the counterfactual votes using observational data,

$$P(V_{ij}^t(g_{ij}^t)|B_{ij}^t) = P(V_{ij}^t|M_{ij}^t = m_{ij}^t, D_{ij}^t = d_{ij}^t, B_{ij}^t),$$

where $V_{ij}^t, M_{ij}^t, D_{ij}^t, R_{ij}^t, B_{ij}^t$ are all observable user behaviors and response features.

To estimate the counterfactual vote estimates, we posit a model of positive voting, following Lee et al. (2016):

$$V_{ij}^t|G_{ij}^t \sim \text{Bern}(\text{sigmoid}(q_{ij}^t + \lambda R_{ij}^t + \nu_i L_{ij}^t + \beta(\frac{1}{1+D_{ij}^t})) \quad (2)$$

for all $i, j, t$, based on the observational data. We fit the model by maximizing the log-likelihood with $l2$-regularization:

$$\max_{\theta} \log \prod_{t=2}^{T} \text{logit}^{-1}(q_{ij}^t + \lambda R_{ij}^t + \nu_i L_{ij}^t + \beta(\frac{1}{1+D_{ij}^t})) - \frac{1}{2}||\theta||_2^2,$$

where $\theta = (\{q_{ij}\}, \lambda, \{\nu_i\}, \beta)$. After training, we are able to get community-level parameters, including the coefficients $\lambda$ and $\beta$; and question level parameter $\nu_i$; and answer level parameter - the response quality of each answer $q_{ij}^t$. We finally produce the response quality estimate:

$$\hat{Q}_{ij}^t \triangleq \sum_{t=1}^{T} \int \mathbb{E}[V_{ij}^t(\tilde{g}_{ij}^t)|B_{ij}^t] \times P(\tilde{m}_{ij}^t, \tilde{d}_{ij}^t) \text{d}\tilde{m}_{ij}^t \text{d}\tilde{d}_{ij}^t,$$

where $\tilde{g}_{ij}^t = (B_{ij}^t, \tilde{m}_{ij}^t, \tilde{d}_{ij}^t)$.

This estimate performs *counterfactual voting adjustment* since it integrates out the existing vote distribution $\tilde{m}_{ij}^t$ and display rank $\tilde{d}_{ij}^t$—wiping out their impacts on the quality estimate—while fixing the response text. This estimator is similar to the backdoor adjustment estimator (or regression adjustment estimator) (Pearl, 2010). It turns out this estimator is the optimal estimator that satisfies counterfactual invariance to existing voting distribution and display rank while minimizing the KL divergence to the original votes under the product distribution. (We expand on this discussion in Appendix A).

**Measurement of position bias and herding bias.** Given the learned parameters for each community, we can quantify the sensitivities to position and herding bias. The displayed-rank related term in the model is $\beta(\frac{1}{1+D_{ij}^t})$. So $\beta$ reflects the extent of the position bias effect: $\beta$ close to 0 implies insensitivity to position bias.

To quantify the herding bias, we estimate the odds between the probability of agreeing with the majority opinion and the probability of disagreeing with the majority opinion via geometric mean across all time steps and for all questions. Further details about the computation of the degree of herding bias are in Appendix B.

## 4. Preliminary Experiments

While the Chinese Voting Process (CVP) accounts for position and herding biases in its generative process (Lee et al., 2016), it lacks an explicit mechanism to mitigate position bias, thereby learning its model parameters for position and herding biases independently. In contrast, our Counterfactual Voting Adjustment (CVA) mitigates both biases by leveraging voting trajectories and backdoor adjustment, jointly learning its model parameters without any independence assumption. This section demonstrates how CVA contextualizes individual votes to infer information quality under each bias.

### 4.1. Mitigating the position bias

To alleviate position bias, votes cast at higher ranks (easy votes) should be down-weighted, while those at lower ranks (hard votes) should be up-weighted. Consider the position bias examples 1a and 1b in Table 1. We assume that two answers exist for the same question: answer A is consistently ranked higher than answer B. In 1a, both answers receive 3 positive votes, but since Answer A gains these votes earlier, it remains ranked higher than Answer B. Their voting sequences are identical, meaning that herding bias is fixed. Their key difference is that all votes for A occur at rank 1, whereas those for B occur at rank 2. Since votes at lower ranks are harder to obtain, up-weighting their importance causes Answer B to be evaluated more favorably than Answer A. The symmetric example 1b replaces positive votes with negative ones, leading to Answer B being assessed more negatively than Answer A.

We learn $q_A^T$ and $q_B^T$ by training them together. Following the convention of CVP, we dropped the first vote of each answer from the training data as it is an arbitrary vote given no context in our setting. So, the quality of each answer at the time of its first vote cannot be estimated. We trained 4 times using the data before $T$ each time. Because this toy example is not real, there's no relative length data involved. The result of the first toy example for position bias is in (a) of Figure 3. Although the total vote of both answers is 3, at the end time, the quality of answer B (0.486) is greater than the quality of answer A (0.466). The result of the second toy example for position bias is in the (b) of Figure 3.

Table 1: **Preliminary Experiment Settings:** Examples 1a and 1b assume two answers of a question where A ranks above B. Both receive identical sequences of three positive (1a) or negative (1b) votes. Example 2 assumes two equally ranked answers of a question having the same vote score (0) but different vote sequences.

| | Example 1a: for position bias | | | | | | Example 1b: for position bias | | | | | | Example 2: for herding bias | | | | | |
|---|---|---|---|---|---|---|---|---|---|---|---|---|---|---|---|---|---|---|
| **Time t** | 1 | 2 | 3 | 4 | 5 | 6 | 1 | 2 | 3 | 4 | 5 | 6 | **Time t** | 1 | 2 | 3 | 4 | 5 | 6 |
| **Answer A (rank 1)** | + | + | + | | | | | | | − | − | − | **Answer A** | + | + | + | − | − | − |
| **Answer B (rank 2)** | | | | + | + | + | − | − | − | | | | **Answer B** | + | − | + | − | + | − |

(a) Learned quality over time for Example 1a   (b) Learned quality over time for Example 1b   (c) Learned quality over time for Example 2

Figure 3: **Preliminary Experiment Results:** Subfigures (a) and (b) demonstrate that CVA mitigates position bias by upweighting the votes received at a lower-ranked position. By time step 6, the learned quality of answer B is more positive than that of answer A in (a), and answer B's learned quality is more negative than answer A's in (b). Subfigure (c) illustrates that CVA mitigates Herding bias by upweighting votes that go against the prevailing majority opinion. At time step 6, the learned quality of answer A is lower than that of answer B.

Although the total vote of both answers is -3, the quality of answer B (-0.486) is more negative than the quality of answer A (-0.467) at the end time. These results satisfy our expectations, and the position bias has been mitigated.

### 4.2. Mitigating the herding bias

To reduce herding bias, votes that follow the majority should be down-weighted, while those opposing the majority should be up-weighted. Consider Example 2 for herding bias in Table 1, where two answers are displayed at the same rank. Answer A receives three consecutive positive votes followed by three negative votes, while Answer B gains alternating votes. Although both answers have a net vote difference score of zero, their vote trajectories differ, implying different quality assessments. For Answer A, its second and third positive votes reinforce the majority and should be down-weighted, while the subsequent negative votes counter the prevailing trend and should be up-weighted, making its final quality more negative. Answer B, experiencing a less dominant majority effect, should have a slightly higher quality than Answer A.

We fix the rank of answers A and B both as rank 1 to learn $q_A^T$ and $q_B^T$ separately. We also omitted the first vote of each answer due to its arbitrariness. As shown in Figure 3(c), Answer A's quality initially rises with positive votes but drops sharply with the first negative vote, reflecting a strong shift against the majority. Answer B's quality fluctuates with alternating votes but remains less negative than Answer A. At the final timestamp, Answer A (-0.146) is more negative than Answer B (-0.118), confirming that herding bias has been mitigated.

## 5. Experimental Results

In our study, we introduce and implement our **CVA** algorithm. After training the model with a customized neural network, we performed rigorous testing on both semi-synthetic dataset and the real StackExchagne dataset, which has been an anonymized compilation of all user-contributed content across the StackExchange network since its launch in 2010. We adopt the dataset version published on October 5, 2022. The network consists of 180 Q&A communities and their corresponding META forums, where users discuss policies and logistics pertaining to their respective communities. The preprocessing details are in Appendix C. After preprocessing, we do the experiments on the largest 120 communities. The statistics of 8 representative communities of StackExchange are in Table 2. "reactjs(SOF)" is a sub-community of StackOverflow about "reacjs."

### 5.1. Semi-synthetic experiment

We generate a semi-synthetic dataset using the proposed model, with detailed data generation steps provided in Appendix D. Using the dataset's ground truth qualities, we

Table 2: **Result Summary** of 8 Representative Communities: In most cases, CVA's quality-based ranking aligns more closely with the true quality ranking than CVP or vote-difference ranking, as measured by Kendall's Tau rank correlation (KT, higher is better) and Sum of Squared Residuals (Res, lower is better).

| Comm | Statistic | | | | | | Semi-Synthetic | | | Real (comment sentiment) | | | Real (GPT-4o helpfulness) | | |
|---|---|---|---|---|---|---|---|---|---|---|---|---|---|---|---|
| | #Questions | #Answers | #Votes | #Comments | | | voteDiff | CVP | CVA | voteDiff | CVP | CVA | voteDiff | CVP | CVA |
| reactjs(SOF) | 2,445 | 10,301 | 212,015 | 15,607 | | KT | 0.3035 | 0.4089 | **0.4106**\*\*\*\* | 0.0383 | 0.1301 | **0.1524**\*\*\*\* | 0.0805 | 0.1575 | **0.1819**\*\*\* |
| | | | | | | Res | 17015 | 13458 | **13364**\*\*\*\* | 7936 | 6716 | **6509**\*\*\*\* | 7514 | 6551 | **6295**\*\*\*\* |
| askubuntu | 1,733 | 6,551 | 101,799 | 57,584 | | KT | 0.2620 | 0.3603 | **0.3691**\*\*\*\* | 0.0266 | 0.0950 | **0.1180**\*\* | 0.0580 | 0.1116 | **0.1823**\*\*\* |
| | | | | | | Res | 10987 | 9337 | **9098**\*\*\*\* | 6769 | 6245 | **6222**\*\* | 6631 | 6122 | **5753**\*\*\*\* |
| mathoverflow | 998 | 8,733 | 119,383 | 49,002 | | KT | 0.2774 | 0.3759 | **0.3902**\*\*\*\* | 0.1034 | 0.0579 | **0.1160** | 0.1055 | 0.0276 | **0.1495**\* |
| | | | | | | Res | 10438 | 8232 | **8024**\*\*\*\* | 8280 | 8528 | **7940**\* | 8408 | 8961 | **7679**\*\*\* |
| cstheory | 325 | 2,460 | 33,098 | 3,808 | | KT | 0.3909 | 0.4640 | **0.4659**\*\* | 0.0788 | 0.1275 | **0.1317** | 0.1464 | 0.1936 | **0.2037** |
| | | | | | | Res | 2306 | 1924 | **1892**\*\*\*\* | 2032 | 1872 | **1856**\* | 1926 | 1766 | **1735**\* |
| politics | 1,161 | 6,629 | 113,378 | 31,121 | | KT | 0.3779 | 0.4831 | **0.4880**\*\*\*\* | 0.0253 | 0.1470 | **0.1748**\*\*\*\* | 0.0824 | 0.2138 | **0.2573**\*\*\*\* |
| | | | | | | Res | 7118 | 5485 | **5359**\*\*\*\* | 9691 | 8102 | **7775**\*\*\*\* | 9114 | 7463 | **6865**\*\*\*\* |
| philosophy | 1,590 | 5,991 | 39,079 | 20,481 | | KT | 0.3484 | 0.3473 | **0.3567** | 0.0668 | 0.0768 | **0.0898** | 0.1475 | 0.1530 | **0.1654** |
| | | | | | | Res | 6822 | 6643 | **6566** | 6239 | 6069 | **6067** | **5622** | 5813 | 5652 |
| math.meta | 271 | 2,959 | 46,158 | 13,056 | | KT | 0.4469 | 0.5321 | **0.5360**\*\*\* | 0.1052 | 0.1356 | **0.1411** | 0.1259 | 0.1538 | **0.1775** |
| | | | | | | Res | 2253 | 1869 | **1866**\*\*\*\* | 3831 | **3465** | 3499\*\* | 3764 | 3466 | **3425**\*\* |
| codegolf.meta | 173 | 2,878 | 38,421 | 13,713 | | KT | 0.4837 | 0.5633 | **0.5640**\*\* | 0.1664 | 0.1075 | **0.1070**\* | 0.1797 | **0.1509** | 0.1567 |
| | | | | | | Res | 1950 | 1675 | **1671**\*\*\* | 3843 | 3453 | **3450**\*\* | 3713 | 3430 | **3396** |

significance level of CVA better than voteDiff:    \*(p $\leq$ 0.05)    \*\*(p $\leq$ 0.01)    \*\*\*(p$\leq$ 0.001)    \*\*\*\*(p$\leq$ 0.0001)

Figure 4: Validation on **semi-synthetic** data (Politics community): Dots represent answers. X-axis: true quality rank; Y-axis: rank by vote difference score (left), CVP learned quality (middle), or CVA learned quality (right). Closer alignment to the diagonal indicates better performance—CVA shows the strongest alignment.

assess whether our model can recover a ranking closely aligned with the ground truth quality-based ranking.

**Residual to diagonal line comparison.** For each response (answer), we rank based on system vote difference scores, model-learned quality, and predefined true quality, then normalize ranks into z-scores for comparability across questions. A better ranking shows a stronger correlation with the true ranking. We visualize answers as scatter points, with the x-axis representing true rank z-scores and the y-axis representing the model's rank z-scores. The association is measured using the sum of squared residuals from the diagonal line (x = y), as the ideal ranking aligns perfectly with the true ranking. The results for the synthetic data of the politics community are shown in Figure 4. A smaller residual to the diagonal line indicates better performance. Our proposed model achieves the lowest sum of squared residuals, demonstrating a significant improvement over both the

system vote difference-based ranking and the CVP-learned quality ranking. Additional example community plots are provided in Appendix E.

**Rank Correlation comparison.** For each question, we have rankings of its answers using different models. We compute the Kendall's $\tau$ coefficient (rank correlation) between a model's ranking and the true ranking. Then, average it over questions. The results for synthetic data of 8 representative communities are shown in Table 2. Our model outperforms both the system vote difference ranking algorithm (voteDiff) and CVP for all 8 communities.

### 5.2. Real experiment

Since the ground truth qualities of StackExchange answers are unknown, we use two proxies. The first is comment sentiment, as writing comments requires more effort than

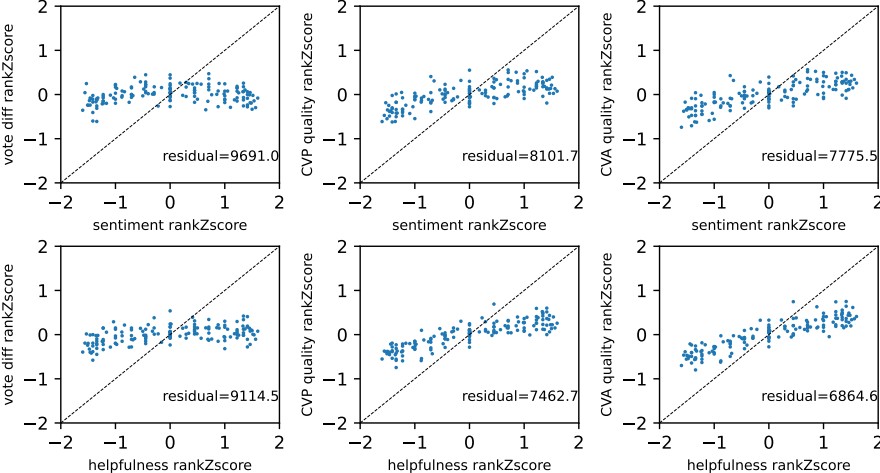

Figure 5: Validation on **real** data (Politics community): Dots represent answers. X-axis (first row): comment sentiment as true quality; X-axis (second row): GPT estimated helpfulness score as true quality; Y-axis: similar to Figure 4. CVA (right) shows the best alignment to the diagonal line with the least residuals.

voting and reflects independent user opinions, making it less biased and closer to true quality. Additionally, Stack-Exhange hired moderators to review the comments and hide trivial comments since 2016. The quality of the comments is reliable. The second is a helpfulness evaluation by GPT-4o, which assesses every single answer in a zero-shot manner given its questions and comments. Previous study (Kamalloo & Rafiei, 2023) found that GPT-4 achieves state-of-the-art on an established benchmark NQ-OPEN for open-domain question-answering tasks, and a zero-shot prompting method can be a reasonable substitute for human evaluation. Given GPT-4o's advancements, we consider it a reliable proxy for ground truth. The prompt example for GPT-4o evaluation is in Appendix F.

Similarly, as for semi-synthetic experiments, we demonstrate our CVA's advantage with less residual to the diagonal line and higher Kendall $\tau$ rank correlation with true ranking.

**Comment Sentiment as ground truth.** The results for real data from the politics community are shown in Figure 5 (first row), with different y-axes: vote difference-based ranking (left), CVP-learned quality (middle), and CVA-learned quality (right). Our model achieves the lowest sum of squared residuals (7776) compared to CVP (8102) and voteDiff (9691). Across all 8 representative communities except math.meta's Res, our model performs best as shown in Table 2. A possible reason is that some meta communities suffer much fewer position biases, so the proposed CVA didn't show its advantage over CVP, but still better than vote difference score. Kendall's $\tau$ rank correlation results in Table 2 also show that CVA consistently achieves the highest correlation across all 8 communities.

**GPT-4o evaluated Helpfulness as ground truth.** The

results for real data from the politics community are in Figure 5 (second row). Our model achieves the lowest sum of squared residuals (6865) compared to CVP (7463) and voteDiff (9115). Across all 8 representative communities except codegolf.meta's KT and philosophy's Res, our model performs the best. Additional plots are in Appendix G.

**Community-level winrate.** As shown in Table 3, among the largest 120 communities, there are 94 (78.3%) communities whose CVA ranking has a smaller residual than both system vote difference-based ranking and CVP quality-based ranking; and there are 89 (74.2%) communities whose CVA ranking has higher KT than both system vote difference based ranking and CVP quality based ranking when using sentiment of comments as ground truth. There are 93 (77.5%) communities' residuals performing the best, and there are 90 (75%) communities' KT performing the best when using the helpfulness score evaluated by GPT-4o as ground truth.

Table 3: Three types of win rates for the largest 120 communities with real data. The entries are the percentages of communities showing that CVA is better than vote difference score (left), CVA is better than CVP (middle), and CVA is better than both (right)

| | | CVA better than vote Diff | CVA better than CVP | CVA better than **both** |
|---|---|---|---|---|
| sentiment of comments as ground truth | Res | 84.17% | 90.00% | **78.33%** |
| | KT | 79.17% | 86.67% | **74.17%** |
| helpfulness score of GPT as ground truth | Res | 81.67% | 94.17% | **77.50%** |
| | KT | 80.83% | 91.67% | **75.0%** |

**Advantages over CVP.** While the proposed CVA does not

always significantly outperform CVP, it offers three key advantages. Firstly, CVP paper didn't provide causal effect estimation. Secondly, CVA is able to mitigate position and herding bias at the same time, while CVP can't. Thirdly, CVA is better than CVP in evaluations, especially for the communities with high position bias, since CVP doesn't consider the rank position when predicting the vote. For those communities with much less position bias, the advantage of CVA won't be shown significantly.

## 6. Analysis and Discussion

In this section, we extensively apply our framework to analyze user voting behavior across 120 communities for broader generalization. Additionally, we present a case study illustrating how our CVA framework addresses the two key counterfactual questions.

### 6.1. Cross-community analysis

We train the largest 120 communities in the StackExchange with the data till October 2022. For 17 communities with enormous data sizes like StackOverflow and MathOverflow, we randomly sampled partial questions due to the limitation of our computational resources. With the learned parameters in Section 3, we compute the sensitivities to position bias and the degrees of herding bias separately for each community and map these two behavioral coefficients into the 2D map as shown in Figure 1. The MEDIAN point is the median value of the measurements of two biases. It divides the map into 4 quadrants. The communities in similar regions share similar voting behaviors and tendencies. Detailed analyses can be found in the caption of Figure 1.

### 6.2. Revisiting the key counterfactual questions

To answer "What if the information were presented at different ranks?", we compute a bunch of probabilities of receiving a positive vote ($P(V^+)$) given different ranks using Equation (2), since we learned the quality and coefficients. The descending speed of $P(V^+)$ from top rank to lower rank reflects the position bias. Similarly, to answer "What if the information had received even votes?", we force the vote ratio feature as even votes (previous positive vote count = negative vote count) to show a non-herding biased $P(V^+)$ for each rank (blue bars). For any response at a certain rank, the $P(V^+)$ in the positive mood (previous positive votes are more than negative votes as in red bars) should be higher than that in the neutral mood (blue bars). Vice versa, the $P(V^+)$ in the negative mood (green bars) should be lower than that in the neutral mood. These differences reflect the herding bias. We average the $P(V^+)$ across all the answers in a community. 'b_realVoteRatio_posMood', 'b_realVoteRatio_negMood' and 'b_evenVoteRatio' are fitted parameters of simplified power-law function as Equation (3). The value

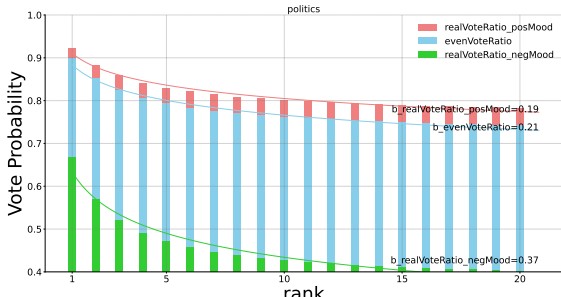

Figure 6: Predict positive vote probabilities given different vote ratios and ranks (**politics** community). Probabilities are higher when the majority opinion is positive (red) than neutral (blue) or negative (green), and decrease with lower ranks.

of $b$ also reflects the position bias. Figure 6 is for politics community. See other community figures in Appendix H.

$$P(V^+) = \frac{1}{rank^b + 1} + c \qquad (3)$$

## 7. Conclusion

Helpfulness votes are often less accurate due to position bias and herding bias. Our **CVA** framework mitigates these biases by accounting for the sequence of votes and underlying causal effects, enabling fairer estimation of response quality. Compared to system vote score-based rankings and the CVP model, CVA 's quality-based rankings show stronger alignment with true quality proxies: comment sentiments and GPT-4o helpfulness evaluations.

This study offers several managerial insights. Platforms can improve answer rankings by recovering true quality, turning high-quality question–answer pairs into valuable knowledge assets. Better rankings reduce users' effort in finding useful content and help recognize contributors whose answers are underappreciated by biased votes.

CVA is adaptable to other platforms, such as product or service review sites, as long as vote history is available. The framework can also integrate richer presentational features, such as images, code, or author reputation.

This research has limitations, mainly due to assumptions made for backdoor adjustment. We assume that perceived answer quality affects only the next vote, not prior ones, and that all relevant confounders are included. Cases where answers were read but not voted on are excluded due to unavailable reading data. However, the framework can be extended to address these issues as more data becomes available. Future work will explore these aspects and aim to bridge the gap between GPT-4o and human evaluation.

## Impact Statement

This paper presents work whose goal is to advance the field of Machine Learning. There are many potential societal consequences of our work, none of which we feel must be specifically highlighted here.

## Acknowledgments

YW was supported in part by the Office of Naval Research under grant N00014-23-1-2590, the National Science Foundation under grant No. 2231174, No. 2310831, No. 2428059, No. 2435696, No. 2440954, and a Michigan Institute for Data Science Propelling Original Data Science (PODS) grant.

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

## A. CVA: Inferring response quality from helpfulness votes

Given the counterfactual vote estimate, how can we form an estimate of the response quality? One criterion is that the predicted response quality $\hat{Q}_{ij}^t(\tilde{g}_{ij}^t)$ be invariant to the extrinsic factors:

$$\hat{Q}_{ij}^t(\tilde{m}_{ij}^t, \tilde{d}_{ij}^t)|G_{ij}^t \stackrel{d}{=} \hat{Q}_{ij}^t(\tilde{m'}_{ij}^t, \tilde{d'}_{ij}^t)|G_{ij}^t, \tag{4}$$

where $\stackrel{d}{=}$ indicates equal in distribution. Moreover, the predicted response quality shall recover the users' voting behavior as much as possible.

To this end, we find the optimal prediction that is closest to the mean of $V_{ij}^t(\tilde{g}_{ij}^t)$ while satisfying counterfactual invariance:

$$\hat{Q}_{ij}^T \triangleq \sum_{t=1}^T \int \mathbb{E}[V_{ij}^t(\tilde{g}_{ij}^t)|B_{ij}^t]P(\tilde{m}_{ij}^t, \tilde{d}_{ij}^t)\mathrm{d}\tilde{m}_{ij}^t\mathrm{d}\tilde{d}_{ij}^t, \tag{5}$$

where $\tilde{g}_{ij}^t = (B_{ij}^t, \tilde{m}_{ij}^t, \tilde{d}_{ij}^t)$. $P(\tilde{m}_{ij}^t, \tilde{d}_{ij}^t)$ is the population distribution of the existing votes and display rank at time $t$ for the $j$th response **across all questions**. The optimality of predicted response quality is due to Theorem 1 of (Wang et al., 2023), while following the same distribution what presentation configurations $\tilde{m}_{ij}^t, \tilde{d}_{ij}^t$ the response was assigned to. Notation-wise, $T$ is the total number of voting actions, so $\hat{Q}_{ij}^T$ calculates the total sum of **average potential** positive votes if the users were shown random configurations of past voting history and at random display rank.

Finally, one can estimate potential voting behavior $\hat{Q}_{ij}^t$, following the fitted parametric voting behavior model:

$$\int \mathbb{E}[V_{ij}^t(\tilde{g}_{ij}^t)|B_{ij}^t]P(\tilde{m}_{ij}^t, \tilde{d}_{ij}^t)\mathrm{d}\tilde{m}_{ij}^t\mathrm{d}\tilde{d}_{ij}^t = E_{\tilde{R}_{ij}^t, \tilde{L}_i^t, \tilde{D}_{ij}^t}[\mathrm{sigmoid}(q_{ij}^t + \lambda R_{ij}^t + \nu_i L_{ij}^t + \beta(\frac{1}{1+D_{ij}^t}))],$$

where one plugs in the estimate of $\lambda, \beta, \nu_i, q_{ij}^t$ from parametric model fit. The expectation is taken over all items (i.e., the population distribution of the voting trajectories at time t for the $j$th response). This is the *CVA* estimate.

## B. Computation of the degree of herding bias

As given in Section 3, the sensitivity to herding bias can be computed as follows:

$$\text{degree of herding bias} = \{\prod_i \prod_{t_i} \frac{P_{ij}^{(t_i)}(agree\_majority\_vote)}{P_{ij}^{(t_i)}(against\_majority\_vote)}\}^{1/n}. \tag{9}$$

The annotation $i$ in the equation indicates a question, and the time tick has a subscript because it's a relative time associated with each question. $n$ is the total number of votes. The ratios of agree majority vote probability over against majority vote probability at all time ticks for all questions are summarized by geometric mean. To generalize this equation using the probability of up-vote $P(vote+)$ at a time tick and the probability of down-vote $P(vote-)$ at a time tick, the degree of herding bias can be computed as below:

$$\{\prod_i \prod_t (\frac{P(v_{ij}^{t+1}=1|q_{ij}^t, \lambda^t, \nu_i^t, \beta^t)}{P(v_{ij}^{t+1}=0|q_{ij}^t, \lambda^t, \nu_i^t, \beta^t)})^{h^{(t_i)}}\}^{1/n} \quad \text{where } h^{(t_i)} = \begin{cases} 1 & \text{when } n_{ij}^{+(:t_i)} \geq n_{ij}^{-(:t_i)}, \\ -1 & \text{when } n_{ij}^{-(:t_i)} > n_{ij}^{+(:t_i)}. \end{cases} \tag{10}$$

The annotation $n_{ij}^{+(:t)}$ means the number of positive votes (up-vote) given to the $j^{th}$ answer of question $i$ before time tick $t$; and $n_{ij}^{-(:t)}$ is the number of negative votes (down-vote) given to the $j^{th}$ answer of question $i$ before time $t$.

The dynamic probabilities of up-vote and down-vote overtime from the whole community view are acquired as below:

$$P(v_{ij}^{t+1}|q_{ij}^t, \lambda^t, \nu_i^t, \beta^t) = \mathrm{sigmoid}(q_{ij}^t + \lambda^t R_{ij}^t + \nu_i^t L_{ij}^t + \beta^t(\frac{1}{1+D_{ij}^t})). \tag{11}$$

## C. Data pre-processing

For real data experiments, we extract data from Posts, Votes, and PostHistory for each community. Among 8 types of Posts, we primarily focus on two main types: "Question" and "Answer". For each community, we extract all the questions and compile them chronologically with respect to their creation time. We exclude the questions that have ever been closed or locked because closed questions are no longer answerable, and locked questions cannot be modified. Then, for each question, we gather all corresponding answers in their chronological order of creation. But we filter out questions with less than 5 answers, aside from the answer accepted by the question author.

For each question, we then organize all votes cast for its answers according to the timestamp of each vote, assigning each vote a relative time index to facilitate cross-question comparisons and integrations. This processing results in the reconstruction of the matrix for each question, whose dimensions are the number of answers by the length of time. The matrix entry is the vote: up-vote as $1$, down-vote as $-1$. We remove all the votes given to the accepted answers after the answers get accepted. This is because the user interface of StackExchange with an accepted answer has changed multiple times. In addition, the fixed top placement of the accepted answer could inject noisy information. Further, an answer is accepted by the question's author when they feel it is good enough. However, It's not necessarily the best answer all the time. Finally, we discard communities with fewer than 100 questions.

## D. Semi-synthetic data generation steps

We generate the same number of questions as the real data and the same number of events, including new answer creation and vote events. At each simulating time, we first choose a question according to the distribution of event counts of each question in real data. Secondly, we decide whether to create a new answer or vote based on the Chinese Restaurant Process (CRP) of one parameter case, and the parameter is computed based on the real data. Next, if choosing to create a new answer, sample a true quality from a normal distribution and use the same relative length as in real data. If choosing to vote, we select an existing answer according to the probability, which is the inverse of the displayed rank. At last, we decide to vote positive or negative using the proposed model and the learned coefficients from real data.

# E. Other example community plots to validate the learned quality on semi-synthetic data

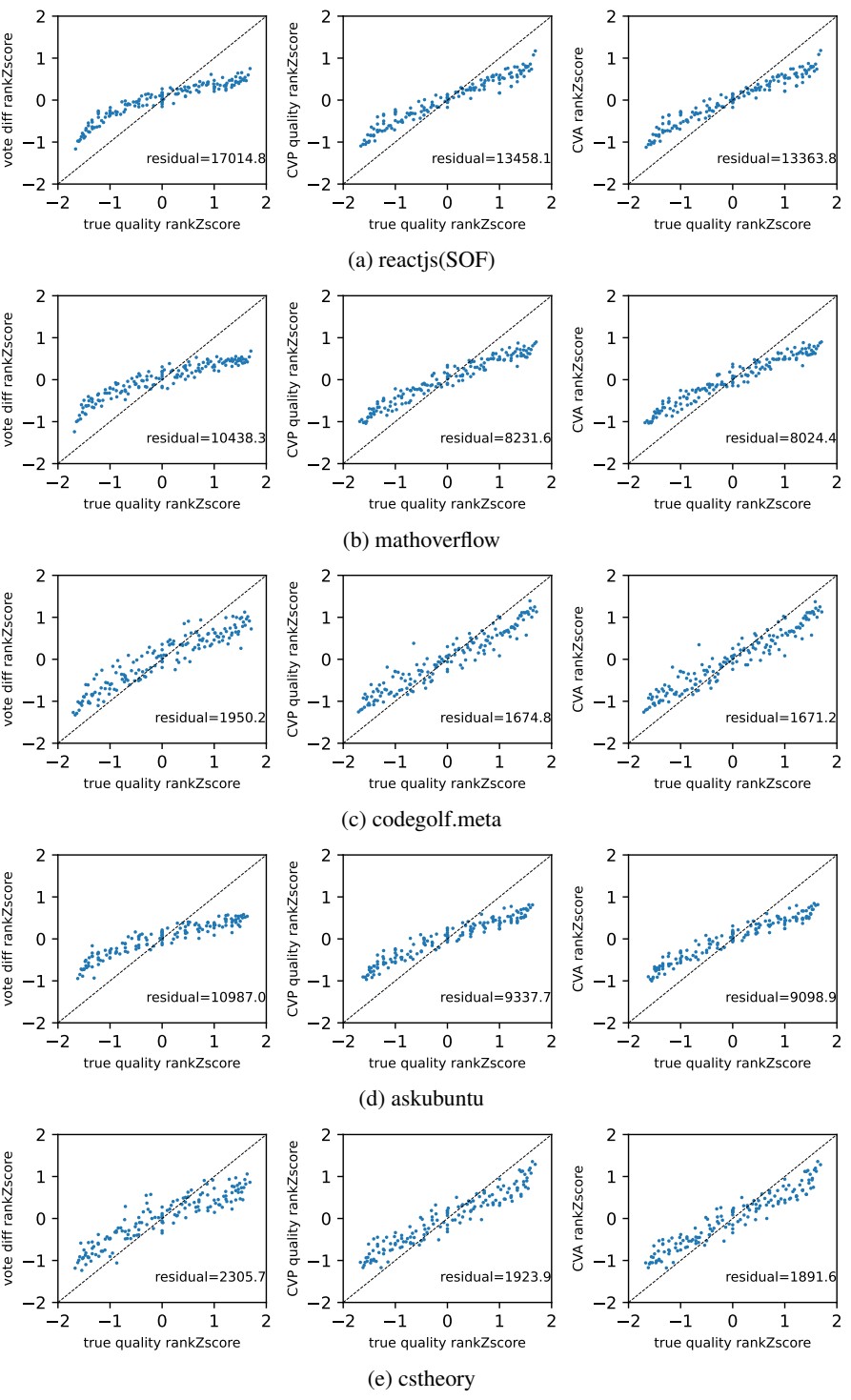

(a) reactjs(SOF)

(b) mathoverflow

(c) codegolf.meta

(d) askubuntu

(e) cstheory

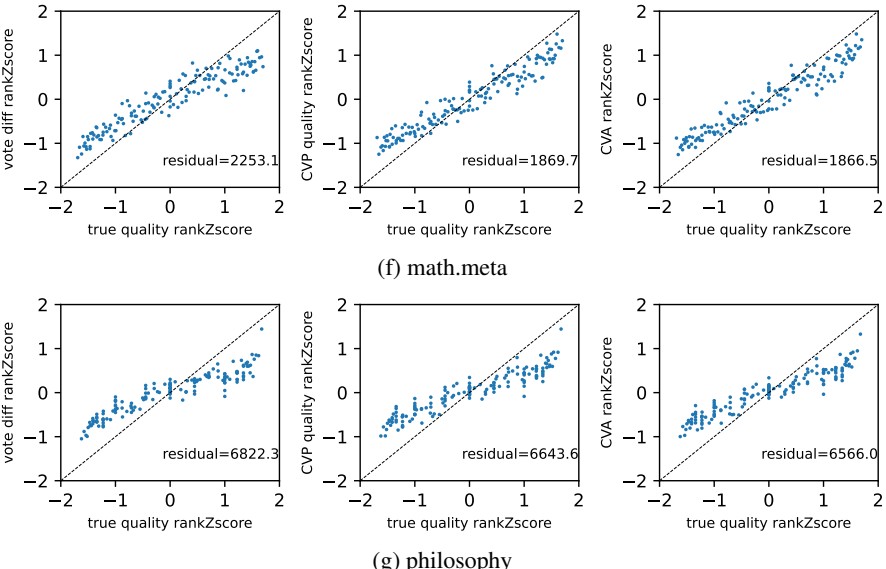

(f) math.meta

(g) philosophy

## F. Prompt template for GPT-4o

For each answer, we ask GPT-4o with the following prompt:

Given the following [QUESTION] and [ANSWER] with 4 COMMENTS from [COMMENT-1] to [COMMENT-4], Do the following 2 tasks.

(1) Estimate a sentiment score from -1 to 1 for each comment. 1 as the most positive and -1 as the most negative. Response in a new line only with the sentiment scores and separate them with commas, such as "0.1,-0.3,0.9".
(2) Estimate a score from -1 to 1 for the [ANSWER] about how helpful it is to the [QUESTION] considering all the COMMENTS. 1 as the most helpful and -1 as the most non-helpful. Response in a new line only with the helpfulness score, such as "0.5".

*[QUESTION]:
Can someone suggest a good book for teaching myself about Lie groups? I study algebraic geometry and commutative algebra, and I like lots of examples. Thanks.

*[ANSWER]:
I like Humphreys' book, Introduction to Lie Algebras and Representation Theory, which is short and sweet, but doesn't really talk about Lie groups (just Lie algebras). I also sometimes find myself looking through Knapp's Lie Groups: Beyond an Introduction. If the material was covered in the Spring 2006 Lie groups course at Berkeley, then I prefer the presentation in this guy's notes.

*[COMMENT-1]:
Before the 2006 course, there was Allen Knutson's 2001 course, from which there are several sets of notes, e.g. URL

*[COMMENT-2]:
Also, Theo Johnson-Freyd has some notes from Mark Haiman's Fall 2008 course here: URL

*[COMMENT-3]:
Finding a reasonably elementary book on Lie groups with lots of examples is challenging. What makes the subject attractive is that it's the crossroads for many subjects. My book definitely wasn't about Lie groups (and has too few examples) but does get somewhat into "modern" representation theory. Knapp is reliable but somewhat advanced. Fulton-Harris is also not a Lie group book and doesn't introduce infinite dimensional representations, but covers a lot of concrete classical examples plus symmetric groups. Free online notes can be a safe starting point, but shop around.

*[COMMENT-4]:
@Anton You and Theo should be both be very proud of your TeX-ed notes, particularly on Lie theory.

# G. Other example community plots to validate the learned quality on real data

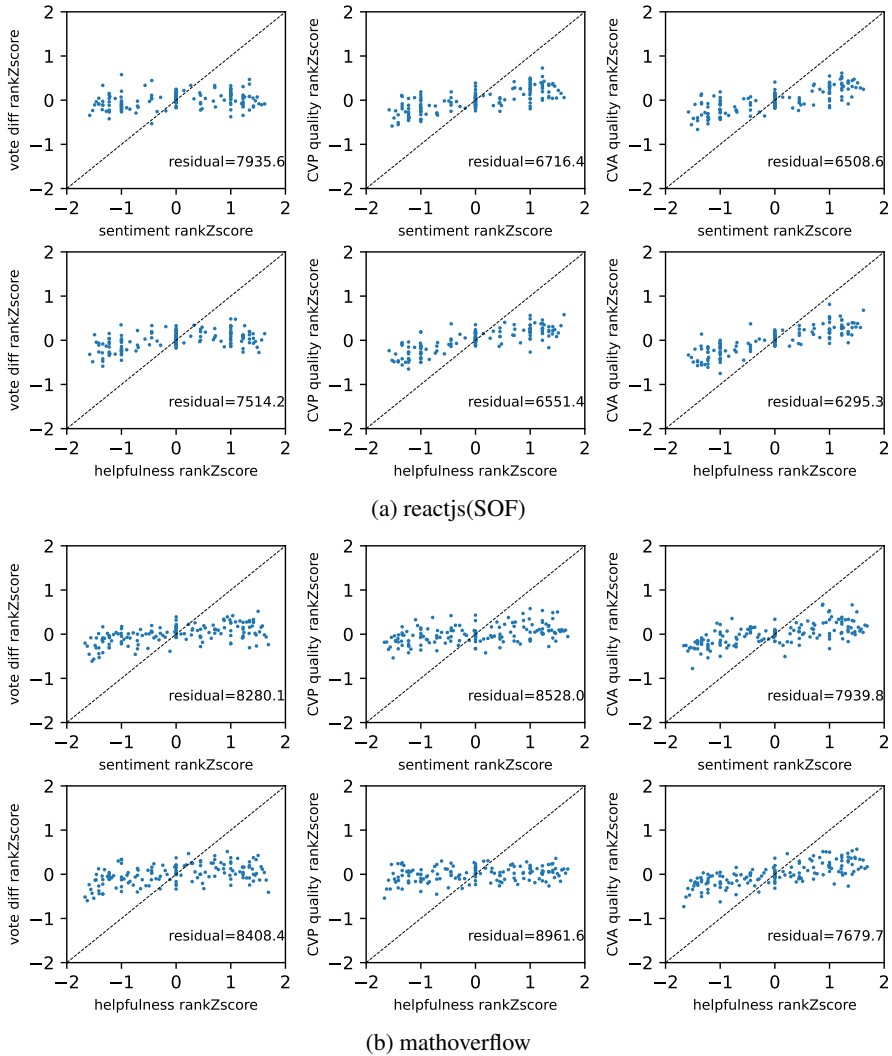

(a) reactjs(SOF)

(b) mathoverflow

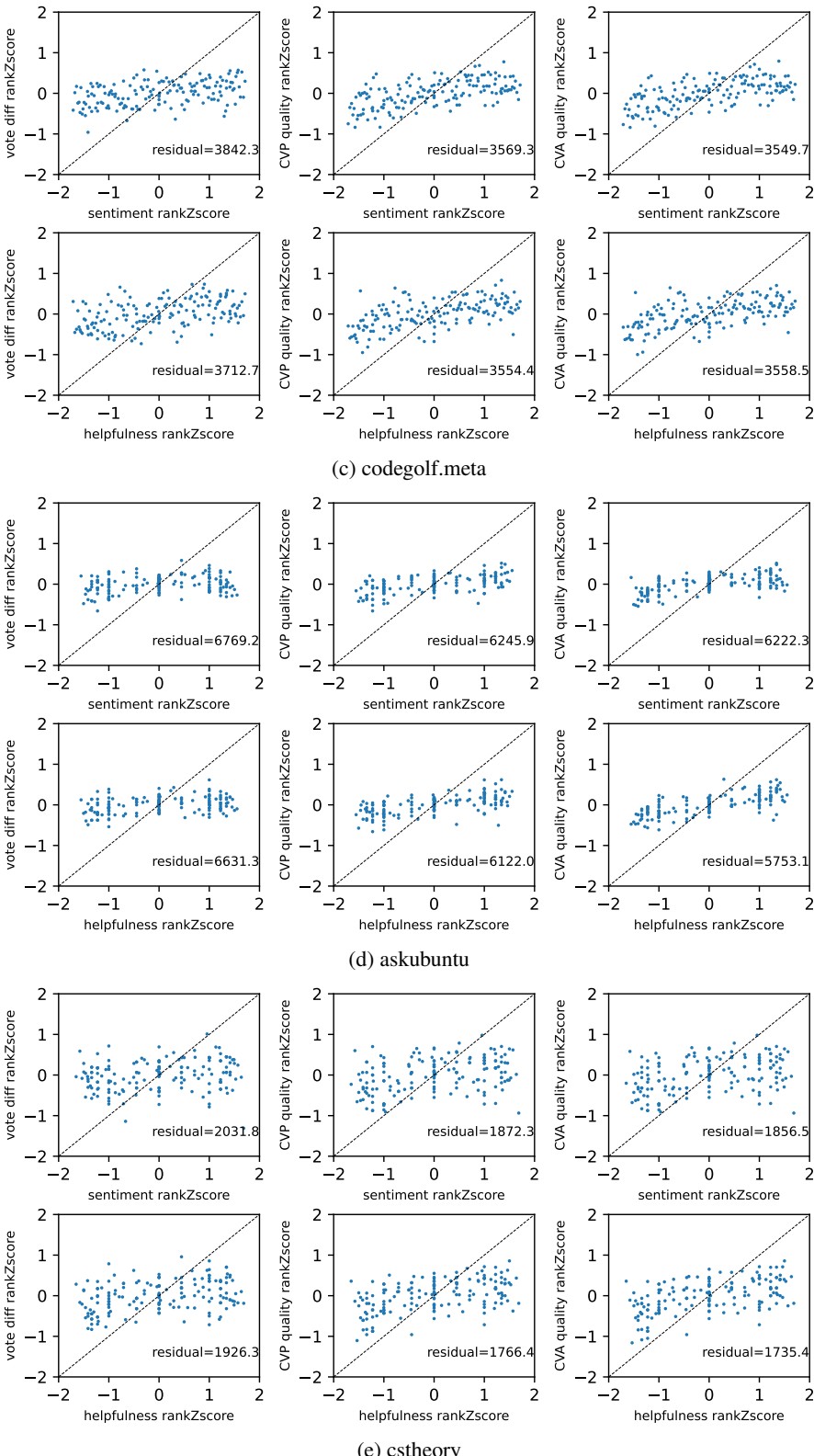

(c) codegolf.meta

(d) askubuntu

(e) cstheory

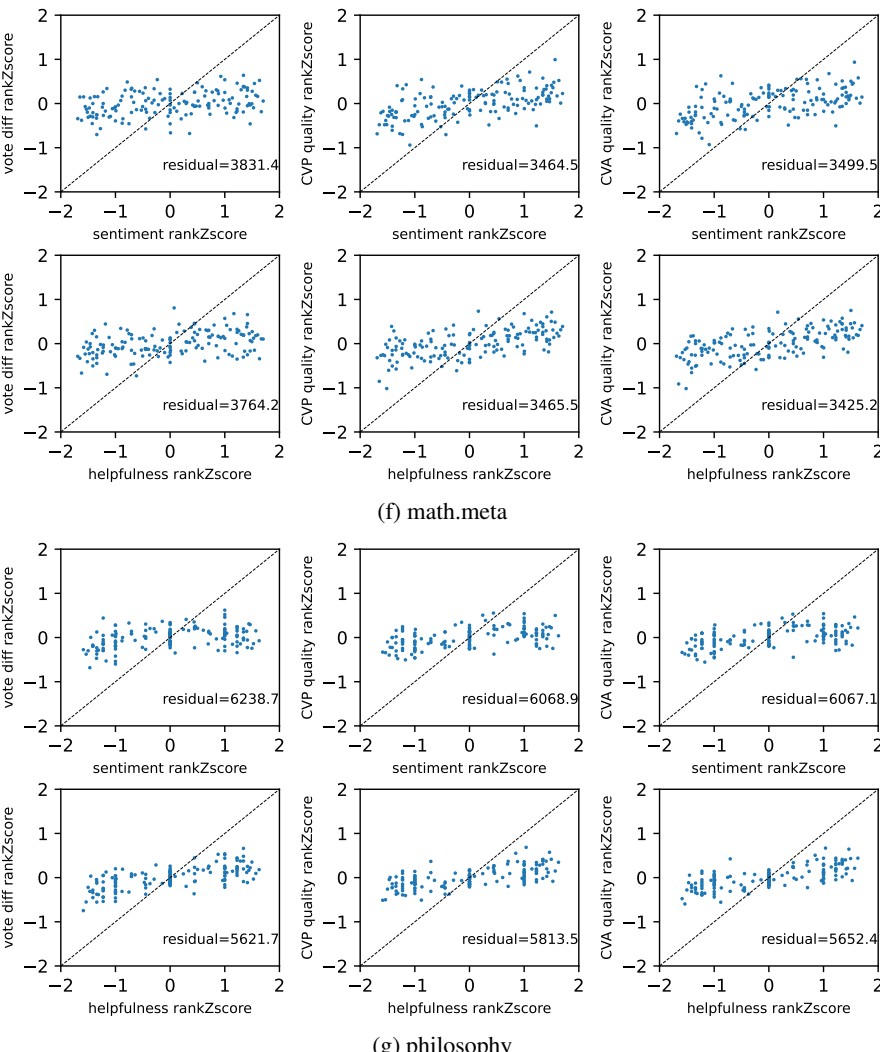

(f) math.meta

(g) philosophy

# H. Other example community plots to answer the counterfactual questions

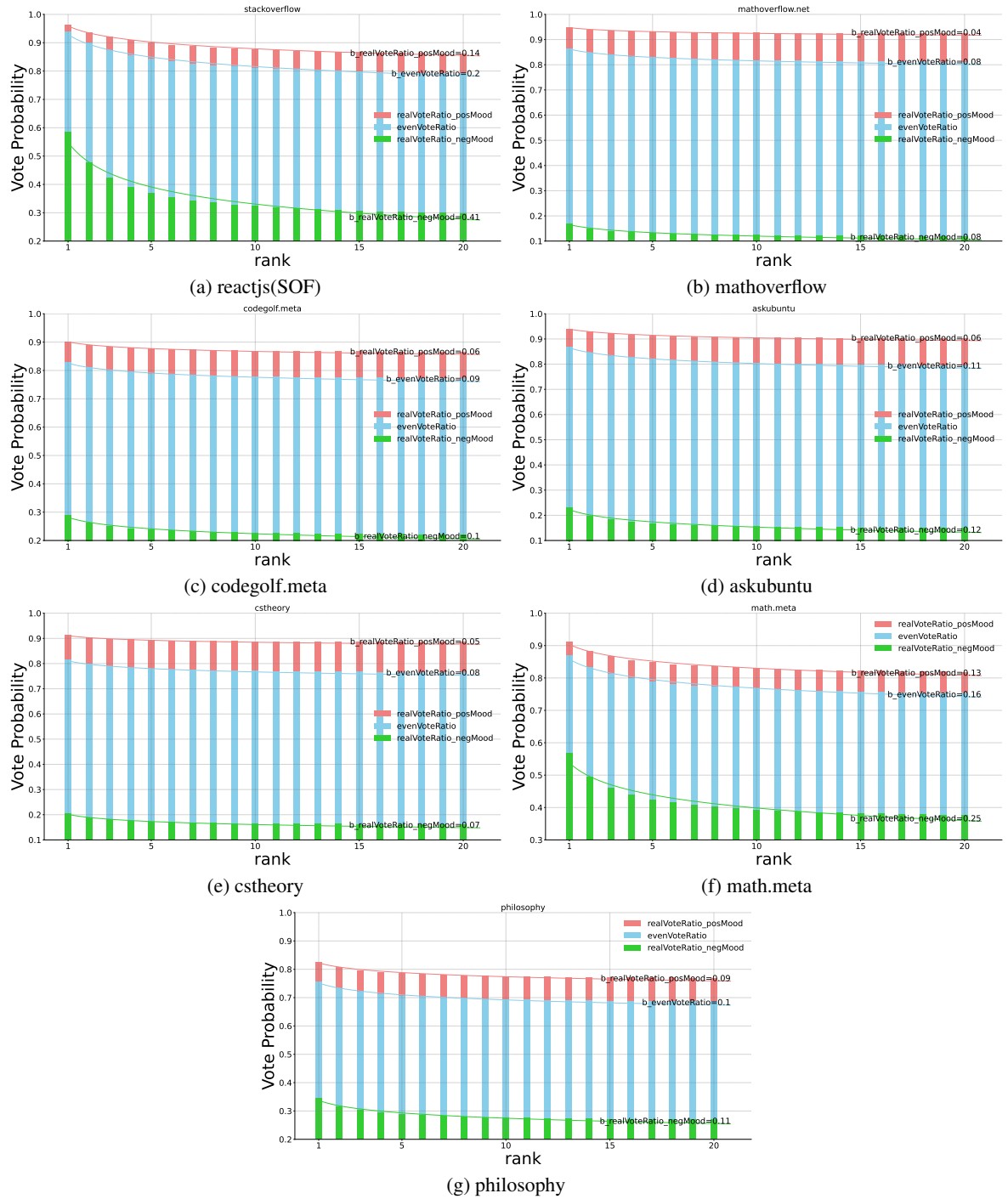

(a) reactjs(SOF)

(b) mathoverflow

(c) codegolf.meta

(d) askubuntu

(e) cstheory

(f) math.meta

(g) philosophy

