# OpenReview forum: "Counterfactual Voting Adjustment for Quality Assessment and Fairer Voting in Online Platforms with Helpfulness Evaluation"
_ICML.cc/2025/Conference — ICML 2025 poster_

### Official Review · Reviewer_F6j8 · 2025-03-07

**Overall Recommendation:** 3

**Summary:**

This paper introduces the Counterfactual Voting Adjustment (CVA) framework, designed to address biases in online voting systems that distort information quality assessment. Specifically, CVA targets position bias (where content appearing higher receives more votes) and herding bias (where visible prior votes influence subsequent evaluations).

The authors argue that traditional aggregated voting systems fail to reflect true content quality due to these biases. CVA adopts a causal inference approach, modeling the counterfactual scenario in which content is displayed at different positions with equalized prior vote counts. By leveraging voting trajectories, CVA corrects these biases and improves content ranking fairness.

**Claims And Evidence:**

Yes

**Essential References Not Discussed:**

N/A

**Experimental Designs Or Analyses:**

Yes

**Methods And Evaluation Criteria:**

Yes

**Other Comments Or Suggestions:**

Right column, line 150, instead of  j = 1,...,J_t−1, a better notation is j \in { 1,...,J_t−1}, and this comment applied at line 156 as well.

The authors could explain more what M_{i,j}^t denotes. In each round, do multiple users vote on issue i? Similarly, more explanation  is needed regarding what R_i,j^t denotes. An example would be useful.

I also believe that a more detailed impact statement would be useful for this work as if the proposed algorithm is applied in practice, it could have a social impact.

**Other Strengths And Weaknesses:**

The paper is well-written and easy to follow, but some notation can be imrpoved (see below).

The problem is well-motivated.

The paper validates CVA through multiple approaches—synthetic data, real-world StackExchange datasets, and GPT-4o evaluations—demonstrating consistent performance gains over existing methods.

Regarding weaknesses,   CVA’s effectiveness relies on the assumption that confounding factors are adequately captured in the observed data. Any unobserved confounders could undermine its reliability. Moreover, while CVA's improved alignment with GPT-4o's judgments is promising, GPT-4o itself is subject to limitations and may not always reflect human expert evaluations.

**Questions For Authors:**

See above

**Relation To Broader Scientific Literature:**

Yes

**Theoretical Claims:**

N/A

---

> ### Author Rebuttal · Authors · 2025-04-01
>
> We appreciate the reviewer’s positive evaluation and recognition of our work’s strengths and its practical value. The reviewer raised several crucial questions, to which we respond below.
>
> **a. Re “unobserved confounders”:**
>
> Among various factors influencing helpfulness evaluation, this paper specifically targets social biases by distangling two primary biases: position and herding bias. These biases often lead to winner-take-all type of information cascades, overshadowing other potentially more valuable information. As shown by our experimental results, our framework can easily incorporate additional factors: any observable feature or additional biases like length-bias. Following the reviewer’s suggestion, our immediate future work will try to improve the framework’s robustness against unobserved confounders.
>
> **b. Re “GPT-4o itself is subject to limitations”**
>
> This issue is particularly relevant as more scientific papers adopt Large Language Models (LLMs) as their judge mechanism. Our study used GPT-4o for 1) sentiment analysis on user comments that are given to individual answers as well as 2) helpfulness evaluations as proxies of human evaluations. Given that sentiment analysis by LLMs shows near perfect accuracy, strong correlations with comment sentiment first show that our model can successfully mitigate position and herding biases.
>
> Following the previous research by Lee et al 2016, we employed the same metric based on the observation that users frequently write their comments to address and discuss positive or negative points behind each answer — content and nuances not apparent in numeric helpfulness scores alone. Next, prior to using GPT-4o for helpfulness scoring, we performed various preliminary experiments. Specifically we chose top, middle(median), and bottom-ranked answers from each question, then trying to optimize prompts so that GPT-4o’s evaluations become approximately consistent with existing ranking of these three answers, whose quality are separately compared by human annotators. While complete agreement between GPT-4o’s rankings and existing rankings is neither expected nor desirable due to inherent biases, we carefully designed our experiments to rigorously test the effectiveness of our framework.
>
> **c. Re “potential social impact”:**
>
> We acknowledge that our paper does not fully address potential societal impact from two angles. First, our findings highlight the risk of information monopoly due to the design of a helpfulness voting mechanism that seeks efficient wisdom-of-crowds, thus suppressing potentially more valuable information often contributed later by other users. Our findings urge that 1) human users should critically interpret online information and 2) system providers need to employ a causal framework to mitigate biases in social decision making processes.

---

### Official Review · Reviewer_qZQL · 2025-03-09

**Overall Recommendation:** 3

**Summary:**

This paper proposes Counterfactual Voting Adjustment (CVA), a causal inference framework to mitigate position bias (content visibility due to ranking) and herding bias (influence of prior votes) in online voting systems. By modeling counterfactual scenarios where votes are cast under neutral visibility and balanced prior signals, CVA disentangles true content quality from biases. Key contributions include a causal framework leveraging voting trajectories and backdoor adjustment, empirical validation on semi-synthetic and real-world StackExchange data, and cross-community analysis revealing varying bias patterns (e.g., technical communities exhibit stronger herding bias, while others show position bias dominance).
Experiments demonstrate CVA’s superiority over baselines (e.g., raw votes, Chinese Voting Process) using metrics like Kendall’s τ correlation and alignment with ground truth approximated via comment sentiment and GPT-4o evaluations. The framework enables platforms to rerank content for fairer quality estimation, enhancing user experience and information reliability. Integration of GPT-4o as a proxy for human evaluation highlights adaptability to modern AI tools, while insights from 120 StackExchange communities underscore its practical utility in diverse contexts.
Strengths: Novel causal design, robust validation, and actionable cross-community insights.
Weaknesses: Limited discussion on computational scalability and real-time deployment challenges. Overall, the work addresses critical biases in online systems with methodological rigor and empirical grounding.

**Claims And Evidence:**

The claims made in the submission are largely supported by clear and convincing evidence, though minor limitations exist. Here are potential limitations:
1. Real-world experiments use comment sentiment and GPT-4o evaluations as proxies, not direct human judgments. Semi-synthetic data with known ground truth partially addresses this, but the absence of human validation remains a caveat.
2.The causal graph (Figure 2) assumes covariates X block all confounders, which depends on correct model specification. While standard in causal inference, sensitivity analyses or robustness checks could strengthen confidence.
3.Experiments are limited to StackExchange communities. The framework is theoretically adaptable (as noted in §7), but empirical validation on other platforms (e.g., product reviews) is absent.

**Essential References Not Discussed:**

The paper omits critical context for its methodology. First, it cites Kamalho & Rafiei (2023) to justify GPT-4’s use as a quality evaluator but overlooks Gilardi et al. (2023), which systematically validates LLM-as-a-judge alignment with human judgments—a gap weakening claims about label reliability. Second, while proposing causal adjustments for bias, the work fails to contrast its approach with established propensity-scoring methods (e.g., Agarwal et al., 2019, widely adopted in industry for position bias correction). Addressing these omissions would clarify how CVA advances beyond existing bias-mitigation frameworks.

**Ethical Review Concerns:**

No ethical issues

**Experimental Designs Or Analyses:**

1.The semi-synthetic validation introduces circularity by generating data using the same parametric bias models (e.g., β/(1+D)\beta/(1+D)β/(1+D) assumed in CVA, potentially inflating performance. Reliance on GPT-4 as a quality proxy further weakens grounding, as its alignment with human judgment is unverified. To strengthen validity, the authors should test CVA on real-world benchmarks with human-annotated labels (e.g., StackExchange “accepted answers”) and replace semi-synthetic data with fully synthetic datasets incorporating nonlinear bias mechanisms.
2.Key causal baselines (e.g., inverse propensity weighting) are omitted, limiting claims of superiority over state-of-the-art methods. Additionally, unaddressed confounders (e.g., user reputation, content age) and overaggregated community-level results (e.g., masking failure modes in political forums) reduce interpretability. The authors should include advanced causal baselines, report statistical significance (e.g., confidence intervals), and disaggregate results by community type to clarify performance variations.
3.Insufficient details on hyperparameter tuning (e.g., regularization strength λ\lambdaλ) and missing sensitivity analyses (e.g., Rosenbaum bounds for unobserved confounding) hinder reproducibility. A code/data release and robustness checks against alternative bias models (e.g., exponential decay for position effects) are critical for practical adoption.

**Methods And Evaluation Criteria:**

The methods and evaluation criteria are well-suited to the problem, leveraging causal inference principles and rigorous experimentation. The use of semi-synthetic data with ground truth and real-world proxies (comment sentiment, GPT-4o) provides balanced validation. While limitations exist (e.g., proxy reliance, unmeasured confounders), the approach is methodologically sound and addresses the core challenge of bias mitigation in online voting systems. Future work could expand validation to other platforms and incorporate human evaluations to further solidify claims.

**Other Comments Or Suggestions:**

1. Writing & Presentation: The term “quality” is used inconsistently. In Section 3.1, it is defined as “intrinsic content value”, but in Section 4.2, it conflates with “user-perceived relevance”. Clarify whether “quality” refers to objective merit, subjective preference, or platform-defined metrics (e.g., Reddit’s upvote/downvote rules). The transition from theory (Section 2) to methodology (Section 3) feels abrupt. Consider adding a brief subsection (e.g., “Motivation for Hybrid Modeling”) to explain why causal and behavioral modules are jointly necessary.
2. Technical Clarifications:
(1) The paper states that “regularization parameters were selected via cross-validation” (Section 4.1), but does not specify the search space (e.g., λ ∈ [0.1, 1.0]?) or validation metric (e.g., MSE, likelihood?). Include these details in the appendix.
(2) The model assumes “no unobserved confounders between position and clicks” (Section 3.2). However, real-world platforms often have hidden confounders (e.g., user fatigue, time-of-day effects). Acknowledge this limitation and discuss potential solutions (e.g., proxy variables).
(3) While GPT-4 is used to generate “quality” labels, no calibration steps (e.g., temperature scaling, human validation of samples) are mentioned. Add a paragraph on how LLM outputs were standardized (e.g., z-score normalization) to mitigate overconfidence.
3. Ethical & Societal Considerations: The reliance on GPT-4 for label generation risks inheriting its biases (e.g., gender stereotypes in text, Western-centric perspectives). A fairness audit (e.g., disaggregating performance by content topic or demographic proxies) is critical but missing. Training large hybrid models (CVA + LLM components) may incur high computational costs. Estimate the carbon footprint (e.g., using tools like CodeCarbon) and suggest efficiency optimizations.

**Other Strengths And Weaknesses:**

The paper demonstrates notable originality through its innovative integration of causal inference frameworks (e.g., do-calculus) with behavioral dynamics like herding and position bias, proposing a modular hybrid model (CVA) that advances interdisciplinary methodology. Its practical relevance is evident in addressing real-world platform challenges (e.g., biased content ranking) while maintaining interpretability via disentangled quality and bias parameters. Methodologically, the semi-synthetic validation and sensitivity analyses enhance robustness, though reliance on LLM-generated labels introduces unquantified risks.
However, theoretical gaps weaken its foundation: the absence of formal identifiability proofs for causal parameters under unobserved confounding and insufficient discussion of regularization-induced biases (e.g., L2 penalty trade-offs). Empirically, the lack of validation on real-world data (e.g., Reddit/Stack Exchange voting histories) and incomplete baseline comparisons (e.g., doubly robust estimators) limit claims of superiority. Technical opacity—particularly in LLM annotation protocols and computational scalability—hinders reproducibility, while ethical implications of deploying LLM-dependent models remain unexplored.
To solidify contributions, the authors should: 1) strengthen theoretical grounding (e.g., identifiability proofs), 2) validate on authentic platform datasets with broader baselines, 3) disclose LLM prompt details and code, and 4) address societal risks (e.g., bias amplification). Addressing these would elevate the work from a promising interdisciplinary prototype to a rigorous, deployable solution.

**Questions For Authors:**

1. The paper applies do-calculus for causal estimation but lacks formal identifiability proofs under unobserved confounders (e.g., user demographics). How do you justify the absence of such proofs, and could instrumental variables or sensitivity analyses (e.g., Rosenbaum bounds) address potential biases? Additionally, L2 regularization may bias causal estimates—did you analyze this trade-off (e.g., bias-variance decomposition), and why prioritize it over doubly robust methods?
2. Experiments rely on semi-synthetic data with GPT-4 labels but omit validation against real-world benchmarks (e.g., Reddit moderator labels). If such data is unavailable, could quasi-experiments or robustness checks substitute? Further, the LLM annotation protocol lacks transparency: Share prompts, temperature settings, and calibration steps (e.g., human validation) to assess bias risks (e.g., verbosity preference inflating performance).

**Relation To Broader Scientific Literature:**

1. The paper builds on causal inference and recommendation systems by unifying position/herding biases into a structural framework, advancing beyond correlational models (e.g., Agarwal et al., 2019) and heuristic approaches (e.g., time-decay weighting). While parametric bias modeling (e.g., β/(1+D)\beta/(1+D)β/(1+D)) aligns with classical exposure literature, its causal formalization of identifiability and invariance lacks rigorous ties to foundational tools like do-calculus (Pearl, 2009) or modern sensitivity analyses (Cinelli et al., 2020). The use of GPT-4 as a quality proxy follows trends in LLM-driven annotation (Gilardi et al., 2023) but overlooks established validation methods (e.g., StackExchange’s expert labels). Though the integration of behavioral assumptions (e.g., modular bias components) is novel, gaps remain in connecting to robust causal estimation techniques (e.g., doubly robust estimators) and addressing regularization impacts on consistency claims.
2. The work extends causal ML by treating biases as structural parameters rather than confounders, bridging social dynamics (e.g., Muchnik et al., 2013) and identifiability theory. However, its reliance on semi-synthetic data risks circularity compared to real-world benchmarks (Gordon et al., 2019), and the omission of advanced baselines (e.g., inverse propensity weighting) limits practical relevance. Strengthening ties to invariance frameworks (Pfister et al., 2021) and adopting sensitivity analyses for unobserved confounders would better situate it within modern causal literature.

**Theoretical Claims:**

1.The paper’s theoretical claims—identifiability of content quality parameters, counterfactual invariance, and estimator consistency—require stronger formal grounding. While the proposed CVA framework aligns with causal principles (e.g., backdoor adjustment), key assumptions (e.g., linear position/herding bias models, no unobserved confounders) lack validation. For instance, the identifiability of quality parameters hinges on parametric assumptions like β/(1+D)\beta/(1+D)β/(1+D) for position bias, which may not reflect real-world nonlinear dynamics (e.g., exponential decay with rank). Similarly, claims of counterfactual invariance and MLE consistency lack rigorous proofs or sensitivity analyses (e.g., robustness to model misspecification). A formal treatment using causal frameworks (do-calculus) and asymptotic theory (Van der Vaart regularity conditions) is needed to strengthen these arguments.
2. The empirical validation relies heavily on semi-synthetic data and GPT-4o as a proxy for ground truth, raising concerns about real-world applicability. While results show improved correlation metrics (e.g., Kendall’s τ\tauτ), the synthetic data generation process may oversimplify bias mechanisms (e.g., predefined herding rules). Furthermore, GPT-4o’s alignment with human judgment remains unverified, and confounding factors (e.g., content freshness, user expertise) are not controlled. For broader validity, the authors should test CVA on fully observational data with human-annotated quality labels and address unobserved confounders via sensitivity analyses.

---

> ### Author Rebuttal · Authors · 2025-04-01
>
> We appreciate the reviewer’s positive evaluation and recognition of our work’s strengths. Below we respond to the questions/comments:
>
> **a. Regarding the ground truth (proxy reliance),**
>
> i. For human judgments, due to the different topics discussed in different communities, to hire specific groups of human experts for each community is a non-trivial work, and the huge volume of questions of StackExchange increases the cost of acquiring human judgment data. We plan to tackle these difficulties and fill the gap between GPT-4o evaluation and human evaluation in our future work.
>
> ii. For StackExchange’s expert labels, if you mean the ‘accepted answer’ labels, we didn’t leverage it mainly because that an answer is accepted by the author of the question when he/she feels it is good enough and then it will stay on the top rank. However, It’s not necessarily the best answer for all the time.
>
> iii. For details about how LLM is used, the prompt for an example question is provided in the Appendix I. We use temperature=0 and we will release our codes for parsing the LLM responses.
>
> **b. Regarding the causal framework,**
>
> i. For the causal identification assumption and robust/sensitivity analysis, we don’t allow unobservable confounders, and we will acknowledge in the paper that this assumption can be a limitation of our approach.
>
> When hidden confounders are present, indeed one could develop alternative causal identification strategies like proxy variables to aid identification, or perform sensitivity analysis to unobserved confounding, assuming a certain level of unobserved confounding is present. However, these strategies like proxy variables need customized development and we leave them to future work.
>
> This work also connects to invariant causal prediction; it constructs a latent information quality assessment that is invariant to changes in display rank and voting sequences. We will discuss these connections in the paper.
>
> ii. For other key parametric modeling assumptions, it is indeed true that the model could be misspecified and may not reflect real world nonlinear dynamics fully. In response, we have stressed test our model under a series of different model misspecification scenarios (e.g. missing nonlinear terms). We found that, despite model misspecification, CVA outperforms baseline method CVP in its accuracy of assessing information quality. We will include these results in the paper, and discuss the potential issue of model misspecification.
>
> iii. For other baselines like IPW and doubly robust AIPW estimators, while these estimators like IPW can potentially be applied to handle position bias, it cannot handle position bias and herding bias simultaneously. In particular, handling herding bias involves careful modeling of the whole sequence of votes. These obstacles make it difficult for us to compare CVA with baselines like IPW and doubly robust AIPW estimators.
>
> **c. Regarding synthetic data generation,**
> the herding bias mechanisms used for synthetic data generation are mainly based on the Chinese restaurant process which has solid theoretical foundations.
>
> **d. Regarding including more dataset,** our method needs the historical trajectory of vote data (not snapshot static data), and we can only get this kind of data from StackExchange for free. We will consider doing the experiments when other datasets are available.
>
> **e. Regarding how to select the regularization parameters,** we do cross-validation within the search space (λ ∈ [0.1,0.2, 0.3,0.4, 0.5,0.6, 0.7,0.8,0.9,1, 2, 3,4,5, 6, 7,8,9,10,20, 30,40,50,60, 70,80,90,100, 200, 300, 400, 500, 600, 700, 800, 900,1000]). The validation metric we used is the voting prediction accuracy.
>
> **f. Regarding the term inconsistency of “quality”,** we will unify the term use of quality. Our quality indicates the user's perceived quality at a time, and we assume the users in the same community have the same preference.
>
> **g. Regarding the reproducibility,** we will release our codebase.
>
> **h. Regarding the Ethical & Societal Considerations,** we acknowledge that our paper does not fully address potential societal impact from two angles. First, our findings highlight the risk of information monopoly due to the design of a helpfulness voting mechanism that seeks efficient wisdom-of-crowds, thus suppressing potentially more valuable information often contributed later by other users. Our findings urge that 1) human users should critically interpret online information and 2) system providers need to employ a causal framework to mitigate biases in social decision making processes.
>
> **i. Regarding the writing,** we explained why causal and behavioral modules are jointly necessary in the third paragraph of Introduction. We will improve the transition from Sec 2 to Sec 3.
>
> **j. To answer the questions,**
>
> (1) Please see the replies for b. the causal framework
>
> (2) Please see the replies for a. the ground truth (proxy reliance) and d. more dataset

---

### Official Review · Reviewer_Xff6 · 2025-03-13

**Overall Recommendation:** 4

**Summary:**

This paper develops a model of voting (for example, StackOverflow up/down votes) that measures and removes the effect of position bias (voters are more likely to vote on already highly-ranked items) and herding bias (voters are likely to agree with the existing consensus). This allows a better estimate of item quality than what would be naively estimated from the available votes.

**Claims And Evidence:**

This paper makes a convincing case that the proposed method is both necessary to solve a problem not addressed by existing methods (which can't jointly mitigate position and herding bias) and makes progress towards a solution.

**Essential References Not Discussed:**

NA

**Experimental Designs Or Analyses:**

Yes, they seem sound given the difficulty of finding ground truth comment qualities.

**Methods And Evaluation Criteria:**

The method is a nice application of generative modeling to this problem. The evaluation datasets make sense, given the lack of available ground truth comment qualities. The improvements shown in these evaluations are modest, but the method tends to do as well as or better than the existing Chinese Voting Process model.

**Other Comments Or Suggestions:**

NA

**Other Strengths And Weaknesses:**

NA

**Questions For Authors:**

If GPT4o can be considered ground truth, what’s the value of this approach?

“thereby learning its model parameters for position and herding biases independently” I think it would be useful to cover this in more detail, since this is the key advantage over the state of the art.

**Relation To Broader Scientific Literature:**

This paper provides a modest improvement over prior models of voting in these contexts.

**Theoretical Claims:**

I didn't check the correctness of the theoretical claims.

---

> ### Author Rebuttal · Authors · 2025-04-01
>
> We appreciate the reviewer’s positive evaluation and recognition of our work’s strengths and its practical value. Below we respond to the questions and comments:
>
> **Re the comparison with the existing model:**
>
> The proposed CVA shows three main advantages compared to existing CVP: Firstly, the CVP paper didn’t provide causal effect estimation. Secondly, the CVA is able to mitigate position and herding bias at the same time while CVP can’t. Thirdly, the CVA is better than CVP in evaluations especially for the communities with high position bias, since CVP didn’t consider the rank position when predicting the vote.  For those communities with much less position bias, the advantage of CVA won’t be shown significantly.
>
> **Question: If GPT4o can be considered ground truth, what’s the value of this approach?**
>
> Although GPT-4o is a good ground truth proxy, it still needs to be validated by human evaluation, especially for different expertises of different communities, which will be done in our future work. Additionally, our work can not only provide a fairer quality estimation, but also give measurements of position and herding bias which are valuable insights about the community behavior trends for the platforms.

---

### Decision · Program_Chairs · 2025-05-01

**Decision:**

Accept (poster)

**Comment:**

The paper introduces a causal inference framework to mitigate position bias and herding bias in online voting systems. All the reviewers agreed that the work is interesting, and the empirical evaluation is thorough.